# View-Driven Multi-View Clustering via Contrastive Double-Learning

**DOI:** 10.3390/e26060470

**Published:** 2024-05-29

**Authors:** Shengcheng Liu, Changming Zhu, Zishi Li, Zhiyuan Yang, Wenjie Gu

**Affiliations:** Information Engineering College, Shanghai Maritime University, Shanghai 201306, China; 202230310065@stu.shmtu.edu.cn (S.L.); 202230310081@stu.shmtu.edu.cn (Z.L.); 202230310055@stu.shmtu.edu.cn (Z.Y.); 202230310066@stu.shmtu.edu.cn (W.G.)

**Keywords:** multi-view clustering, deep learning, contrastive learning

## Abstract

Multi-view clustering requires simultaneous attention to both consistency and the diversity of information between views. Deep learning techniques have shown impressive abilities to learn complex features when working with extensive datasets; however, existing deep multi-view clustering methods often focus only on either consistency information or diversity information, making it difficult to balance both aspects. Therefore, this paper proposes a view-driven multi-view clustering using the contrastive double-learning method (VMC-CD), aiming to generate better clustering results. This method first adopts a view-driven approach to consider information from other views to encourage diversity, thus guiding feature learning. Additionally, it presents the idea of dual contrastive learning to enhance the alignment of views at both the clustering and feature levels. The VMC-CD method’s superiority over various cutting-edge methods is substantiated by experimental findings across three datasets, affirming its effectiveness.

## 1. Introduction

Multi-view data usually include representations from diverse features or sources, where each view contains shared semantic information inherent in the multi-view data. The insights derived from multiple views tend to complement each other [1,2]. Visual information, for instance, can be characterized through diverse techniques like SIFT, HOG, and LBP. Likewise, environmental data like temperature and humidity can be gathered using several sensors positioned across a specified region, while the details of these data might vary, there are overarching similarities in the cluster patterns when viewed from a broader perspective. Multi-view clustering seeks to categorize data into various groups by leveraging insights from all accessible viewpoints [3,4,5,6]. However, acquiring knowledge from multiple sources simultaneously is challenging [7].

Multi-view learning (MVC) has gained significant interest across various machine learning applications, such as feature selection [2], scene recognition [8], and information retrieval [9,10,11]. Traditional machine learning approaches can be broadly classified into four categories: subspace learning methods [12], non-negative matrix factorization (NMF) methods [13,14], graph-based techniques [1,15], and a range of kernel-based methods [16]. However, traditional shallow models often struggle to effectively learn feature representations from large datasets [17].

To tackle these challenges, a plethora of deep learning techniques have been introduced [18,19,20,21,22]. Deep multi-view clustering endeavours to improve performance by harnessing the feature representation capabilities inherent in deep models. In essence, these methods strive to enhance distinct consensus representations by utilizing specialized encoder networks for each viewpoint to transform the data.

Recently, contrastive learning has been integrated into deep learning frameworks to obtain unique representations from various viewpoints [23,24]. Most contrastive learning techniques focus on maximizing the common information found in the distributions of multiple perspectives [25]. As an illustration, Yang et al. [26] utilized existing data as positive samples and randomly chose certain cross-view samples as negative samples in deep multi-view clustering (MVC).

Although these multi-view learning techniques have shown impressive outcomes, they still encounter common challenges. Initially, prior algorithms often map diverse views into a unified space under the assumption that there exists a substantial correlation between views. However, in practice, both correlation (consistency) and independence (complementarity) coexist, making it difficult to balance them. Consequently, current algorithms are either focused on maximizing correlation for coherence [27] or on achieving independence for diversity [28], thereby merging coherence and diversity into an intricate and unified challenge [29,30].

In contrast to previous methods, this paper introduces a VMC-CD methodology. Specifically, in the process of learning different view representations, it considers the interesting information from other views and employs two levels of contrastive learning to restrict them for potential clustering and further guide feature learning. Clustering-level and feature-level contrastive learning are two essential aspects that contribute to distinct stages of feature learning and synergistically strengthen each other. In clustering-level contrastive learning, the invariant representations of multiple views are aligned and feature information obtained from a single view is normalized from the perspective of the clustering assignment. Feature-level contrastive learning endeavours to align encoded feature representations specific to each view, thereby mitigating heterogeneity among different views to some extent. This paper introduces an attention-driven approach to the generation of a discriminative feature representation, which aids in guiding feature learning. This method boosts attention towards information from all views while diminishing the influence of information specific to irrelevant subsets of views regarding clustering. Moreover, this paper illustrates that aligning latent feature distributions across different views using contrastive learning can achieve robust view invariance. Additionally, this approach outperforms existing deep view clustering methods in terms of clustering results, even without the requirement of a well-initialized autoencoder. To summarize, the key contributions of this paper include the following:Introduction of a VMC-CD technique, which incorporates valuable information from alternate views while learning feature representations across diverse viewpoints. It provides guidance information in an attention-driven manner, effectively integrating multiple views into a discriminative common representation to guide feature learning.Introduction of dual contrastive learning, conducting contrastive learning at both the clustering and feature levels, encouraging consistency in clustering across multiple views while preserving their feature diversity.Experiments on three multi-view datasets, demonstrating the effectiveness of the VMC-CD method.

## 2. Related Work

This section delves into recent advancements in pertinent areas, specifically focusing on multi-view clustering and contrastive learning.

### 2.1. Multi-View Clustering

Multi-view clustering and classification techniques can generally be categorized into two main types: conventional methods and deep learning-based approaches. Conventional multi-view clustering methods can be subdivided into five distinct categories. Firstly, some methods are achieved through non-negative matrix factorization techniques, such as that in Liu et al. [31], who explored common latent factors between multiple views and established a deep structure [32] to find more consistent shared features.

Approaches in the second category utilize self-representation to illustrate the relationships among samples [33]. In research conducted by [5], a self-representation layer was employed to hierarchically reconstruct view-specific subspaces and encoding layers, thereby enhancing the consistency of cross-view subspaces.

Approaches in the third category employ dimensionality reduction methods to convert multi-view data into a common, low-dimensional space, enabling a uniform representation. Subsequently, clustering outcomes are derived using established clustering methods [34]. Canonical Correlation Analysis (CCA) [35] is a notable technique within this branch. In a recent study [36], a versatile framework was introduced for reducing the dimensionality of multi-view data, enabling the handling of multi-view feature representations within kernel space.

Methods in the fourth category employ graph models for multi-view clustering [37,38]. The central concept of this method is to identify a common graph among various perspectives and then utilize spectral graph techniques (like spectral clustering) on this shared graph to derive clustering outcomes. Moreover, a study by [39] introduced graph autoencoders for learning multi-view representations. The study [40] focused on extracting valuable insights from complex multi-view data dispersed across various high-dimensional spaces. Through graph learning, the fundamental correlations between different views are uncovered, thereby addressing the issue of effective multi-view collaboration.

The last category of methods tackles this issue using kernel function strategies [41,42], frequently utilizing predefined kernel functions like Gaussian kernels to handle diverse views. Subsequently, these methods linearly or nonlinearly blend these kernel functions to establish a uniform kernel. Yet, the primary challenge with this method is the identification of appropriate kernel functions.

These statistical models face a common limitation in their ability to capture intricate structures within the data. As a result, deep multi-view clustering has garnered considerable attention within the community and has demonstrated effectiveness across various practical scenarios.

In early research, Wang et al. [18] employed a deep autoencoder design to acquire a consolidated representation of multi-view data, yielding commendable results in speech and visual analysis tasks. Subsequently, Andrew et al. [27] introduced an enhancement to Deep Canonical Correlation Analysis (DCCA). Their work centred on creating a unified representation of multi-view data by maximizing the correlation between extracted deep features and CCA. Subsequently, Abavisani et al. [43] introduced a deep multi-view subspace clustering network aimed at revealing a unified affinity matrix across all viewpoints. Moreover, Zhu et al. [44] utilized deep autoencoders for self-representation learning and incorporated diversity and ubiquitous regularization to capture meaningful interconnections among different viewpoints.

While existing algorithms typically prioritize either maximizing view correlation for consistency or maximizing view independence for complementarity, this paper advocates for emphasizing diversity alongside maintaining consistency. This balanced approach aims to achieve improved results by striking a harmonious equilibrium between diversity and consistency.

### 2.2. Contrastive Learning

Contrastive learning has significantly progressed in the realm of self-supervised representation learning [24]. Fundamentally, contrastive learning strives to enhance the feature space of raw data by amplifying similarities among positive pairs (similar instances) while reducing the similarities among negative pairs (dissimilar instances) [45]. Positive pairs generally consist of data from the identical instance, while negative pairs consist of data from different instances.

For instance, Chen et al. [24] introduced a visualization representation framework within contrastive learning. This framework seeks to optimize the agreement between diverse augmented views of a singular example within the latent feature space.

Lately, an approach named Contrastive Prediction (COMPLETER) [46] has advanced significantly by combining reconstruction, cross-view contrastive learning, and cross-view dual prediction methodologies. This method stands out not just for its effectiveness in incomplete multi-view clustering but also for its ability to simultaneously handle data recovery and consistency learning in incomplete multi-view datasets.

These methods contribute to learning high-quality representations based on data. However, determining invariant representations across multiple views remains a challenging problem.

## 3. Methods

In this section, we initially present a clear formulation of the problem and delineate its particulars. Next, we propose a network framework to address this problem. We then delve into each module of the proposed network, including the deep autoencoder module, dual contrastive learning module, and attention weight learning module, in detail.

### 3.1. Problem Formulation

Given a set of multi-view data $X={\left\{X^{\left(\nu \right)}\in R^{d_{v}\times N}\right\}}_{\nu =1}^{n_{v}}$ with $n_{v}$ views and N samples, let us denote the v-th view of the multi-view data as $X^{\left(v\right)}$. Each view $X^{\left(\nu \right)}=\left[X_{1}^{\left(\nu \right)},X_{2}^{\left(\nu \right)},\ldots ,X_{N}^{\left(\nu \right)}\right]$ has N samples, where $X_{i}^{\left(\nu \right)}\left(1\leq i\leq N\right)$ dimensions are $d_{v}$; this represents a certain sample dimension of a particular view, and it is important to note that different samples in each view may have different dimensions. Given K as the cluster count, instances with identical semantic labels can be grouped together into a shared cluster. Hence, there is a requirement to partition N samples into K distinct clusters.

### 3.2. Overview of the Network Architecture

According to Figure 1, the VMC-CD method aims to directly extract semantic labels for end-to-end clustering from raw data instances spanning multiple perspectives. We achieve this by applying the dual contrastive learning module to feature representation learning, introducing an end-to-end deep clustering network structure. Additionally, we have given special treatment to the encoder by integrating a view-driven attention mechanism. As shown in the diagram below, the proposed VMC-CD network architecture consists of three main modules: the deep autoencoder module, the dual contrastive learning module, and the attention weight learning module (AT BLOCK). The core of the entire architecture is the deep autoencoder module, which learns features conducive to clustering across multiple perspectives through unsupervised representation learning. The dual contrastive learning module is divided into two parts: one part performs contrastive learning on the discriminative feature representation learned by the aforementioned encoder–decoder, known as the feature-level contrastive learning part, and the other part optimizes parameters through contrastive soft clustering assignment, known as the clustering-level contrastive learning part. The attention weight learning module primarily enhances the clustering level of the discriminative feature representation by leveraging information from other views.

### 3.3. Deep Autoencoder Module

Our network architecture primarily relies on a deep autoencoder module comprising multi-view feature encoders and multi-view feature decoders. When learning feature representations, our attention mechanism incorporates information from other views. Thus, during the feature encoding phase, we take into account relevant information from alternate views. Illustrated in the diagram below, the multi-view feature encoder utilized in this study comprises two components: a view-specific autoencoder module and an attention module influenced by other views, specifically the attention weight learning module (discussed in detail in Section 3.5).

The view-specific encoder module comprises three initial blocks: a linear layer, a batch normalization layer, and an activation layer (ReLU). The feature encoder module primarily aims to convert view-specific data into a discriminative feature representation. This is achieved by integrating the output of the view-specific autoencoder module with the output of the attention module, which is influenced by information from other perspectives. Subsequently, a softmax function is applied to generate the feature representation. On the other hand, the feature decoder module performs the opposite operation, converting the discriminative feature representation back to the original view information. The construction of each decoder block is the same as that of the encoder block.

The overview steps of the autoencoder module are as follows: First, based on input feature data $X_{i}^{\left(\nu \right)}$, the encoder component acquires a compact representation $Z_{i}^{\left(v\right)}$, where $X_{i}^{\left(\nu \right)}$ represents the i-th sample of the v-th view, and $Z_{i}^{\left(v\right)}$ represents the low-dimensional representation of the i-th sample of the v-th view. This is the simplified formula:(1)$$Z_{i}^{\left(v\right)}=f_{E}\left(X_{i}^{\left(\nu \right)}\right)$$Here, $f_{E}\left(.\right)$ broadly refers to a series of operations by the encoder on the input data $X_{i}^{\left(\nu \right)}$. For the v-th view, where v = m, the specific encoder is as follows:(2)$$r_{mi}^{\left(e\right)}=\left\{\begin{array}{cc} f\left(BN\left(W_{r}^{(e)T}X_{i}^{\left(m\right)}+b_{r}^{\left(e\right)}\right)\right) & e=1\\ f\left(BN\left(W_{r}^{(e)T}r_{mi}^{\left(e-1\right)}+b_{r}^{\left(e\right)}\right)\right) & e=2,3\\ f\left(BN\left(W_{r}^{(e)T}r_{mi}^{\left(e-1\right)}+b_{r}^{\left(e\right)}\right)\right) & e=4 \end{array}\right.$$
$r_{mi}^{\left(e\right)}$ represents the latent representation of the i-th sample of the m-th view after passing through a certain layer of the encoder. $W_{r}^{\left(e\right)}$ denotes the weights, while $b_{r}^{\left(e\right)}$ denotes the biases of the encoder segment. BN represents the batch normalization operation. f(·) represents the ReLU activation function. The attention module driven by other views obtains attention weights through the sigmoid function, denoted by $o_{mi}^{\left(6\right)}$ (details on how to calculate this will be provided in Section 3.5). This module embeds the interesting information into the attention weights, which are then element-wise multiplied with the view-specific feature representation, resulting in a view-specific discriminative representation. The specific formula is as follows: (3)$$Z_{i}^{\left(m\right)}=\zeta \left(r_{mi}^{\left(4\right)}⊗o_{mi}^{\left(6\right)}\right)$$Here, ξ(.) represents the result after passing through the sigmoid function. Subsequently, the decoder transforms the reconstructed features $Z_{i}^{\left(m\right)}$ back into the original input data by extracting hidden representations. The Equation is as follows: (4)$$X_{i}^{\left(m\right)}=f_{D}\left(Z_{i}^{\left(m\right)}\right)$$The specific decoder is as follows: (5)$$r_{mi}^{\left(d\right)}=\left\{\begin{array}{cc} f\left(BN\left(W_{r}^{(d)T}Z_{i}^{\left(m\right)}+b_{r}^{\left(d\right)}\right)\right) & d=1\\ f\left(BN\left(W_{r}^{(d)T}r_{mi}^{\left(d-1\right)}+b_{r}^{\left(d\right)}\right)\right) & d=2,3,4 \end{array}\right.$$
$r_{mi}^{\left(d\right)}$ represents the latent representation of the i-th sample of the m-th view after passing through a certain layer of the decoder. $W_{r}^{\left(d\right)}$ and $b_{r}^{\left(d\right)\;}$ represent the weights and biases of the encoder part. Let $\hat{X_{i}^{\left(m\right)}}=r_{mi}^{\left(4\right)}$. $\hat{X_{i}^{\left(m\right)}}$ represents the reconstructed data of the i-th sample of the m-th view. This allows us to construct the reconstruction loss function. In this study, the autoencoder network’s objective function is attained through the minimization of the reconstruction error. After extrapolating the loss of the m-th view to all views, the total reconstruction loss is as follows: (6)$$\ell_{rec}=\sum_{v=1}^{n_{v}}\sum_{i=1}^{N}∥X_{i}^{\left(v\right)}-\hat{X_{i}^{\left(v\right)}}∥$$

### 3.4. Dual Contrastive Learning Module

The dual contrastive learning module is divided into two parts. One part performs contrastive learning on the discriminative feature representation learned by the encoder–decoder, referred to as feature-level contrastive learning in this paper. The other part optimizes parameters through contrastive clustering assignment, referred to as clustering-level contrastive learning.

Feature-level contrastive learning is performed within the latent space of the autoencoder representation to explore the common information representation across various views. This process focuses on learning the alignment between different views by maximizing their mutual information. The loss function for feature-level contrastive learning is denoted by: (7)$$\ell_{ch}=\sum_{1\leq m\leq n\leq n_{v}}\sum_{i=1}^{N}\left(I\left(Z_{i}^{\left(m\right)},Z_{i}^{\left(n\right)}\right)+\partial \left(H\left(Z_{i}^{\left(m\right)}\right)+H\left(Z_{i}^{\left(n\right)}\right)\right)\right)$$Here, I represents mutual information, H represents entropy, and the parameter *∂* is used to regularize entropy. According to information theory, entropy represents information content. Hence, a higher entropy $H\left(Z_{i}^{\left(m\right)}\right)$ signifies a larger information content within $X_{i}^{\left(m\right)}$, ensuring diversity of information across different views. Additionally, maximizing mutual information $I\left(Z_{i}^{\left(m\right)},Z_{i}^{\left(n\right)}\right)$ between $H\left(Z_{i}^{\left(m\right)}\right)$ and $H\left(Z_{i}^{\left(n\right)}\right)$ will maintain information coherence across diverse perspectives during feature acquisition.

The contrastive clustering assignment used here is a soft clustering assignment method, unlike hard clustering, which allows data points to be assigned to multiple categories with different probabilities or membership degrees. Soft clustering assigns each data point a membership value for every category, denoting the degree to which the data point pertains to that specific category. These membership values form a membership matrix, where data points are rows and categories are columns, reflecting the membership of data points to each category. In contrast, hard clustering assignment requires each data point to be explicitly and uniquely assigned to a single category, without allowing for sharing or ambiguity. The specific application in this paper is as follows: For any view V = v, after obtaining $r_{vi}^{\left(4\right)}$, a separate branch is opened for all views to undergo further processing. The specific operational process is as follows: (8)$$r_{vi}^{\left(e\right)}=W_{r}^{\left(e\right)}r_{vi}^{\left(e-1\right)}+b_{\gamma }^{\left(e\right)}\;e=5,6$$As shown in the above equation, we first pass it through two linear layers with the purpose of dimensionality reduction, making the dimension of this vector equal to the number of clusters K, in order to proceed with the next step of soft clustering assignment. Let $r_{\nu i}^{\left(6\right)}=H_{i}^{\left(\nu \right)}$. If all samples are processed uniformly according to the above, matrix ${\left\{H^{\left(v\right)}\in R^{N\times K}\right\}}_{v=1}^{n_{v}}$ can be obtained. Here, $H_{i}^{\left(v\right)}$ denotes the i-th row of $H^{\left(v\right)}$ and $H_{ij}^{\left(v\right)}$ represents the j-th element of the i-th row of matrix $H^{\left(v\right)}$, indicating the likelihood that sample i in view vs. belongs to cluster j.

To enhance the diversity between cluster assignments and thus strengthen the effectiveness of soft clustering, ${\left\{Q^{\left(v\right)}\in R^{N\times K}\right\}}_{v=1}^{n_{v}}$ is used to reinforce the results of ${\left\{H^{\left(v\right)}\in R^{N\times K}\right\}}_{v=1}^{n_{v}}$, improving the performance of soft clustering. The calculation process for each element in $Q^{\left(v\right)}$ is as follows: (9)$$Q_{ij}^{\left(v\right)}=\frac{{\left(H_{ij}^{\left(v\right)}\right)}^{2}/\sum_{i=1}^{N}H_{ij}^{\left(v\right)}}{\sum_{k=1}^{K}\left({\left(H_{ik}^{\left(v\right)}\right)}^{2}/\sum_{i=1}^{N}H_{ik}^{\left(v\right)}\right)}$$Let $Q_{j}^{\left(\nu \right)}$ be the j-th column of $Q^{\left(v\right)}$. Each element in $Q_{j}^{\left(\nu \right)}$, denoted by $Q_{ij}^{\left(v\right)}$, represents the soft clustering assignment of sample i to cluster j. Therefore, $Q_{j}^{\left(\nu \right)}$ denotes the clustering assignment of samples belonging to the same semantic cluster. Samples that are the same across different views share the same semantic information. The similarity between two clustering assignments $Q_{j}^{\left(v_{1}\right)}$ and $Q_{j}^{\left(v_{2}\right)}$ for cluster j can be measured by the following equation: (10)$$s\left(Q_{j}^{\left(v_{1}\right)},Q_{j}^{\left(v_{2}\right)}\right)={\left(Q_{j}^{\left(v_{1}\right)}\right)}^{T}Q_{j}^{\left(v_{2}\right)}$$The symbols $v_{1}$ and $v_{2}$ represent two different views, but the clustering assignment probabilities of instances between different views are similar because these instances represent the same samples. Additionally, if instances from multiple views are used to describe different samples, they are uncorrelated with each other. The similarity between cluster assignments within clusters should be maximized, while the similarity between cluster assignments across clusters should be minimized. We perform clustering on samples concurrently, ensuring coherence in the clustering assignments. The cross-view contrastive loss between $Q_{k}^{\left(v_{1}\right)}$ and $Q_{k}^{\left(v_{2}\right)}$ is defined as follows:(11)$$\begin{array}{c} \ell^{\left(\nu_{1},\nu_{2}\right)}=-\frac{1}{K}\sum_{k=1}^{K}\log\frac{e^{s\left(Q_{k}^{\left(v_{1}\right)},Q_{k}^{\left(v_{2}\right)}\right)/\tau }}{T}\\ T=\sum_{j=1,j\neq k}^{K}e^{s\left(Q_{j}^{\left(v_{1}\right)},Q_{k}^{\left(v_{2}\right)}\right)/\tau }+\sum_{j=1}^{K}e^{s\left(Q_{j}^{\left(v_{1}\right)},Q_{k}^{\left(v_{2}\right)}\right)/\tau } \end{array}$$

The symbol τ represents a temperature parameter, $\left(Q_{k}^{\left(v_{1}\right)},Q_{k}^{\left(\nu_{2}\right)}\right)$ denotes positive clustering assignments pairs between two views, while $\left(Q_{j}^{\left(v_{1}\right)},Q_{k}^{\left(v_{1}\right)}\right)$ and $\left(Q_{j}^{\left(v_{1}\right)},Q_{k}^{\left(\nu_{2}\right)}\right)$(j≠k) represent negative clustering assignment pairs between two views. The cross-view contrastive loss induced across multiple views is designed as follows:(12)$$\ell_{c}=\frac{1}{2}\sum_{v_{1}=1}^{n_{v}}\sum_{v_{2}=1,v_{2}\neq v_{1}}^{n_{v}}\ell \left(v_{1},v_{2}\right)$$

The cross-view contrastive loss explicitly compares clustering assignment pairs across multiple views. It pulls together pairs from the same cluster assignment and pushes apart pairs from different cluster assignments. To avoid a scenario where all instances are grouped into a single sub-cluster, we introduce a regularization term as follows:(13)$$\ell_{a}=\sum_{\nu =1}^{n_{v}}\sum_{j=1}^{K}P_{j}^{\left(\nu \right)}\log P_{j}^{\left(\nu \right)}$$

The term $P_{j}^{\left(\nu \right)}=\frac{\sum_{i=1}^{N}Q_{ij}^{\left(\nu \right)}}{N}$ represents the loss defined as the cross-view consistency loss, which prevents all instances from belonging to the same cluster ‘j’. Therefore, the total loss for the contrastive clustering level is as follows:(14)$$\ell_{cl}=\ell_{c}+\mu \ell_{a}$$

### 3.5. The Attention Weight Learning Module (AT BLOCK)

As shown in Figure 2, in learning the feature representations of multiple views, attention is generated from other views, incorporating interest information from these views during the feature encoding process.

We constructed the AT BLOCK using fully connected layers with ReLU and a transformer, connecting the transformer with two ReLU fully connected layers’ inputs and outputs through skip connections. The primary purpose of AT BLOCK is to map complex data into spaces corresponding to different views, obtaining attention weights for different views to guide feature learning.

The attention module is structured with a sequence of fully connected layers followed by ReLU activation. Through the sigmoid function, the attention module calculates attention weights, which encapsulate relevant information within the dataset.

In the multi-view feature encoder input, with two views, the feature learning procedure integrates input from the other view’s data into the attention-driven module to support feature learning. This is symbolized as $A_{1}=X_{2}$ and $A_{2}=X_{1}$. The specific process of the attention module is as follows:(15)$$O_{mi}^{\left(e\right)}=\left\{\begin{array}{cc} f\left(W_{o}^{(e)T}A_{i}^{\left(m\right)}+b_{o}^{\left(e\right)}\right) & e=1\\ f\left(W_{o}^{(e)T}O_{mi}^{\left(e-1\right)}+b_{o}^{\left(e\right)}\right) & e=2,4,5\\ t\left(O_{mi}^{\left(e-1\right)}\right) & e=3\\ O_{mi}^{\left(2\right)}+O_{mi}^{\left(5\right)} & e=6 \end{array}\right.$$Here, $A_{i}^{\left(m\right)}$ represents the input to the attention module for the i-th sample of the m-th view, $O_{mi}^{\left(e\right)}$ represents the feature representation after the e-th layer of the attention module, and t(·) represents the result after the transformer block. $W_{o}^{\left(e\right)}$ and $b_{o}^{\left(e\right)}$ denote the weights and biases of the linear layer in the e-th layer.

### 3.6. Total Loss Function

After introducing all the losses and their computation methods, we can obtain the total loss function of VMC-CD: (16)$$\ell_{vmc}=\ell_{cl}+\lambda_{1}\ell_{ch}+\lambda_{2}\ell_{rec}$$Here, $\lambda_{1}$ and $\lambda_{2}$ are weighting hyperparameters. $\ell_{cl}$ represents the loss function for cluster-level contrastive learning. $\ell_{ch}$ represents the loss function for feature-level contrastive learning. $\ell_{rec}$ represents the loss function for reconstruction. The weighted sum of these three constitutes the total loss function $\ell_{vmc}$ in this context.

### 3.7. Complexity Analysis

Let α and β represent the mini-batch size and maximum number of neurons in the proposed network architecture’s hidden layer, respectively. Let $d_{z}$ denote the dimensionality of the view feature representation. The overall complexity of the model is denoted by $O\left(\alpha \beta n_{v}d_{v}\right)$, while the complexities of the reconstruction loss, feature-level contrastive learning loss, and cluster-level contrastive learning loss are represented by $O\left({\alpha n}_{v}d_{v}\right)$, $O\left(\alpha d_{z}n_{v}\right)$ and $O\left(\alpha^{2}Kn_{v}\left(\left(n_{v}-1\right)+n_{v}\left(K-1\right)\right)+n_{v}K\right)$, respectively. Therefore, the overall complexity of the proposed method is denoted by $O\left(T\left(\alpha \beta n_{v}d_{v}+\alpha d_{z}n_{v}+\alpha^{2}K^{2}n_{v}^{2}\right)\right)$, where T represents the maximum number of iterations during training.

### 3.8. Algorithm Flow

This algorithm flow (Algorithm 1) is shown below.
**Algorithm 1 ** View-driven dual-contrastive learning in multi-view clusteringRequirements: multi-view data samples $X={\left\{X^{\left(\nu \right)}\in R^{d_{v}\times N}\right\}}_{\nu =1}^{n_{v}}$, maximum number of iterations $T_{\max}$1: Initialize the parameters of the autoencoder network and set t = 02: While t < $T_{\max}$ and loss function $\ell_{vmc}$ is not converged do3: Compute the loss and update the parameters of the entire network4: t = t + 15: Obtain discriminative feature representations for all views6: Concatenate different view feature representations of the same sample to form $\left[Z^{1}:\ldots :Z^{n_{v}}\right]$, pass it through the k-means clustering algorithm, yielding the clustering outcome denoted as Q7: Output: Clustering result Q

## 4. Experiment

In this section, comprehensive experiments were conducted to evaluate the efficacy of the VMC-CD method proposed in this study. We performed experiments on five commonly utilized multi-view datasets, evaluating the performance of our method against other established multi-view clustering techniques. The source code of VMC-CD is implemented in Python 3.7. All experiments were carried out using a system that includes a GeForce RTX 3080 Ti GPU with 16 GB of memory, a 12th Gen Intel Core i9-12900H CPU, and 32 GB of RAM.

### 4.1. Experimental Setup

#### 4.1.1. Datasets

In this study, we utilized five commonly used datasets. These datasets are Caltech101-20 [47], Scene-15 [48], LandUse-21 [49], MNIST-USPS [50,51] and BDGP [52]. The Caltech101-20 dataset comprises 2386 images representing 20 subjects and incorporates HOG and GIST features as distinct perspectives. The LandUse-21 dataset includes 2100 satellite images across 21 classes, utilizing PHOG, LBP, and GIST features. The Scene-15 dataset comprises 485 images showcasing 15 scenes and incorporates PHOG, LBP, and GIST features. The MNIST-USPS dataset is a handwritten digit image dataset with two different styles, each view containing 10 categories with 500 examples per category. The BDGP dataset has 2500 drosophila embryo images, five categories, each image with 1750-dimensional visual and 79-dimensional textual features for clustering.

#### 4.1.2. Evaluation Metrics

In this study, we utilize accuracy (ACC), normalized mutual information (NMI), and the adjusted Rand index (ARI) as the primary metrics to assess clustering performance. Improved clustering outcomes are indicated by higher values on these metrics.

#### 4.1.3. Network Architecture and Parameter Settings

The VMC-CD model was trained using the Adam optimizer with an initial learning rate of 0.0001. The batch size was fixed at 256, and the number of training iterations varied depending on the dataset: 200 iterations for Caltech101-20, MNIST-USPS, and BDGP, 700 iterations for LandUse-21, and 500 iterations for Scene-15. For all datasets, the entropy parameter in the feature-level contrastive learning, denoted as *∂*, is set to 9, while the temperature coefficient, denoted as τ, is set to 1. The hyperparameters $\lambda_{1}$, $\lambda_{2}$, and μ are chosen from the range [0.05, 0.1, 0.2, 0.5, 1] based on different datasets. For cluster-level contrastive learning, two linear layers are established. The dimension of the first linear layer is selected from the range of [32, 64] depending on the dataset, while the dimension of the second linear layer is configured to match the number of clusters in the dataset.

### 4.2. Performance Evaluation

As shown in Table 1, for Caltech101-20, Scene-15, and LandUse-21 datasets, this study contrasted the proposed approach with 11 other multi-view clustering methods, including IMG (Incomplete Multi-View Grouping for Visual Data) [53], EERIMVC (Efficient and Effective Regularized Incomplete Multi-View Clustering) [16], DAIMC (Dual-Aligned Incomplete Multi-View Clustering) [54], UEAF (Unified Embedding Alignment Framework) [55], DCCAE (Deep Canonical Correlation Autoencoder) [56], PVC (Partial Multi-View Clustering) [57], AE2-Nets (Autoencoders in Autoencoder Networks) [58], DCCA (Deep Canonical Correlation Analysis) [27], PICCAE (Probabilistic Incomplete Canonical Correlation Autoencoder) [59], COMPLETER (Contrastive Prediction) [46], and ATTENTION (Attention-Driven Deep Multi-View Clustering) [60].

In comparison with ATTENTION, the proposed method achieved relative improvements in accuracy of 2.82%, 2.84%, and 1.08% on the Caltech101-20 dataset, Scene-15 dataset, and LandUse-21 dataset, respectively.

As shown in Table 2, we also conducted experiments on two additional datasets and compared our model with other models that perform well on these datasets, achieving excellent results. The comparison methods include: Deep Embedded Clustering (DEC) [22], Improved Deep Embedded Clustering (IDEC) [61], Binary Multi-View Clustering (BMVC) [62], Multi-View Clustering via Late Fusion Alignment Maximization (MVC-LFA) [63], Deep Adversarial Multi-View Clustering Network (DAMC) [21], Self-Paced and Auto-Weighted Multi-View Clustering (SAMVC) [64], Cognitive Deep Incomplete Multi-View Clustering Network (CDIMC-net) [65], End-to-End Adversarial-Attention Network for Multi-Modal Clustering (EAMC) [6], and Reconsidering Representation Alignment for Multi-View Clustering (SiMVC and CoMVC) [54].
On the MNIST-USPS dataset, our model outperforms the second-best method, CoMVC [66], by 0.99%, 1.11%, and 0.79% in terms of ACC, NMI, and ARI, respectively. On the BDGP dataset, our model surpasses the second-best model, DAMC [21], by 0.78%, 2.21%, and 3.16% in ACC, NMI, and ARI, respectively.

Therefore, our model was evaluated on a total of five datasets. In previous studies, models typically demonstrated excellent performance on only a limited number of datasets. In contrast, the VMC model exhibited outstanding performance across all five datasets, showcasing its exceptional generalization capabilities. This broad applicability highlights the robustness and versatility of the VMC model.

### 4.3. Ablation Studies

According to the overall loss equation, which includes three different loss components; the first part is the reconstruction loss component for obtaining consensus representation, the second part is the feature-level contrastive learning component, and the third part is the cluster-level contrastive learning component. To validate the importance of components in VMC-CD, we conducted ablation studies using the same experimental settings to isolate external interference factors. Specifically, we considered two special cases: one where only cluster-level loss is considered during end-to-end training, without considering feature-level loss and another where only feature-level loss is considered during end-to-end training, without considering cluster-level loss. The table displays the results of these two special cases along with the three metrics of our model. The clustering outcomes presented in the initial two rows of the table correspond to the two distinct scenarios. As anticipated, optimal performance is attained when both feature-level contrastive learning and cluster-level contrastive learning are simultaneously incorporated.

In terms of accuracy, VMC-CD with dual contrastive learning outperformed the model without feature-level loss by 9.22%, 0.98%, and 0.09% on the three datasets and the NMI and ARI metrics also showed improvements. VMC-CD with dual contrastive learning also performed better compared to the model without cluster-level loss, with improvements of 24.85%, 3.41%, and 2.38% on the three datasets. Therefore, dual contrastive learning plays a crucial role in learning invariant representations across views and is indispensable. The specific experimental results are shown in Table 3, Table 4 and Table 5.

### 4.4. Parameter Sensitivity Analysis

As shown in Figure 3, we conducted experiments on the Caltech101-20 dataset to study the sensitivity of the parameters $\lambda_{1}$ and $\lambda_{2}$ in the proposed VMC-CD method. The $\lambda_{1}$ parameter is selected from {0.1, 0.5, 1, 2}, and the $\lambda_{2}$ parameter is selected from {0.01, 0.05, 0.1, 0.5, 1}. The chart showcases how the VMC-CD method performs in clustering, measured by ACC, NMI, and ARI scores, across various combinations of $\lambda_{1}$ and $\lambda_{2}$. The clustering performance of the VMC-CD method on the Caltech101-20 dataset varies with different combinations of $\lambda_{1}$ and $\lambda_{2}$. The results exhibit relative consistency in ACC and NMI scores but sensitivity to the $\lambda_{1}$ parameter concerning ARI. Specifically, an increase in the $\lambda_{1}$ parameter correlates with a notable decrease in ARI.

### 4.5. Training Analysis

Figure 4 displays the curves depicting the evolution of each clustering metric with respect to the iteration count on the Scene-15 dataset. The illustrated curves showcase the exceptional stability of the method proposed in this paper, consistently delivering robust clustering performance.

Apart from the previously discussed visualizations, we also include t-SNE [67] visualizations showcasing the learning of a unified representation on the Caltech101-20 dataset. As shown in Figure 5, with an increase in the number of epochs, the learned representation becomes more condensed and distinctive.

Additionally, the following Table 6 presents the number of iterations and runtime of the model on five datasets, further illustrating the speed and efficiency of the proposed model in this paper.

## 5. Discussion

The VMC-CD method effectively addresses the challenge of multi-view clustering by balancing consistency and diversity of information. It not only advances multi-view clustering techniques but also aligns with trends in deep learning and data clustering research. Emphasizing a view-driven approach and dual contrastive learning, it improves clustering performance and feature alignment. Future directions may include exploring dynamic dataset handling and high-dimensional data applications. VMC-CD represents significant progress in multi-view clustering, inspiring research in deep learning and data clustering.

## 6. Conclusions

This paper introduces a view-driven dual-contrastive learning approach for multi-view clustering. This method involves incorporating relevant information from other views during the feature representation learning phase, promoting view diversity, and facilitating consensus feature learning. The concept of dual-contrastive learning is introduced, which promotes view consistency from both the clustering level and the feature level, complementing each other.

## Figures and Tables

**Figure 1 entropy-26-00470-f001:**
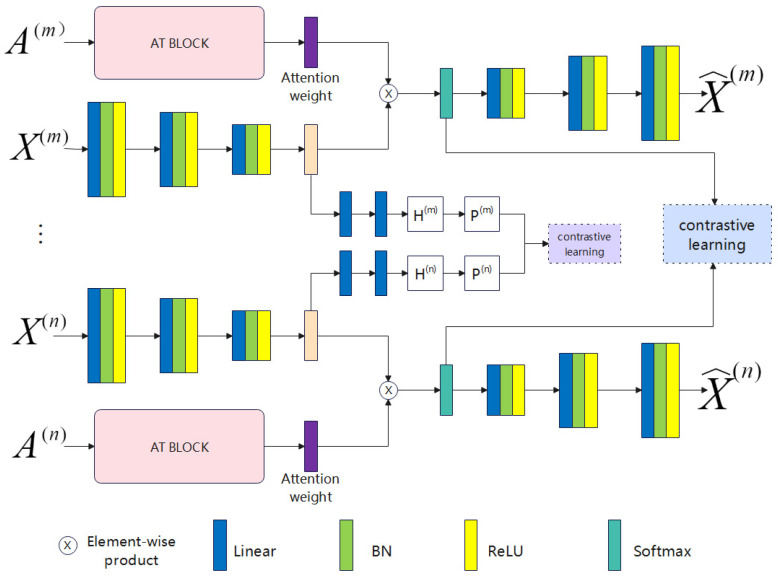
Architecture of the view-driven dual contrastive learning for multi-view clustering.

**Figure 2 entropy-26-00470-f002:**
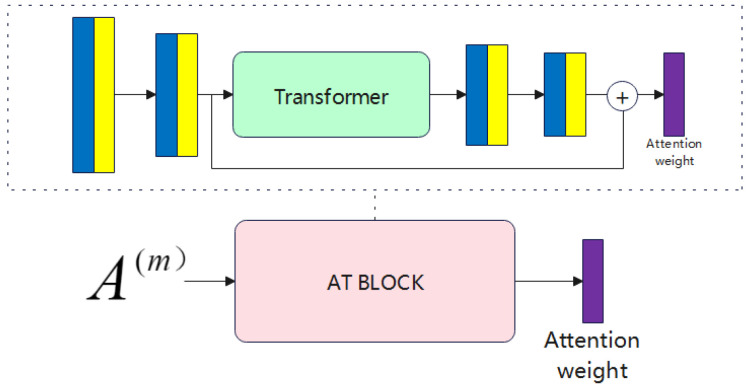
Architecture of the attention weight learning module.

**Figure 3 entropy-26-00470-f003:**
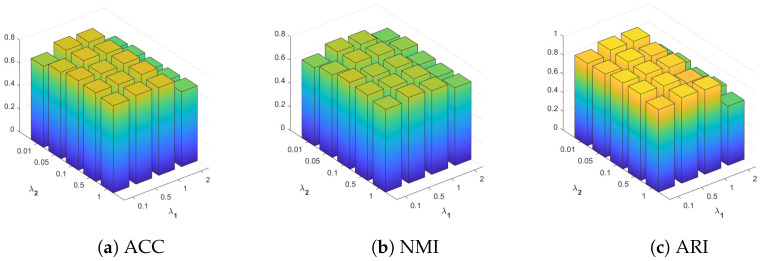
Sensitivity analysis of parameters for Caltech101-20.

**Figure 4 entropy-26-00470-f004:**
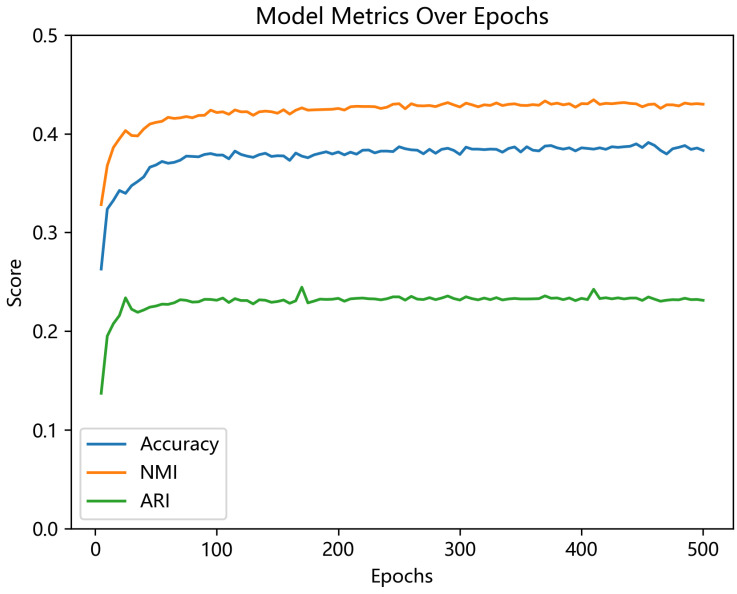
Change curve of clustering metrics with iteration count on Scene-15 dataset.

**Figure 5 entropy-26-00470-f005:**
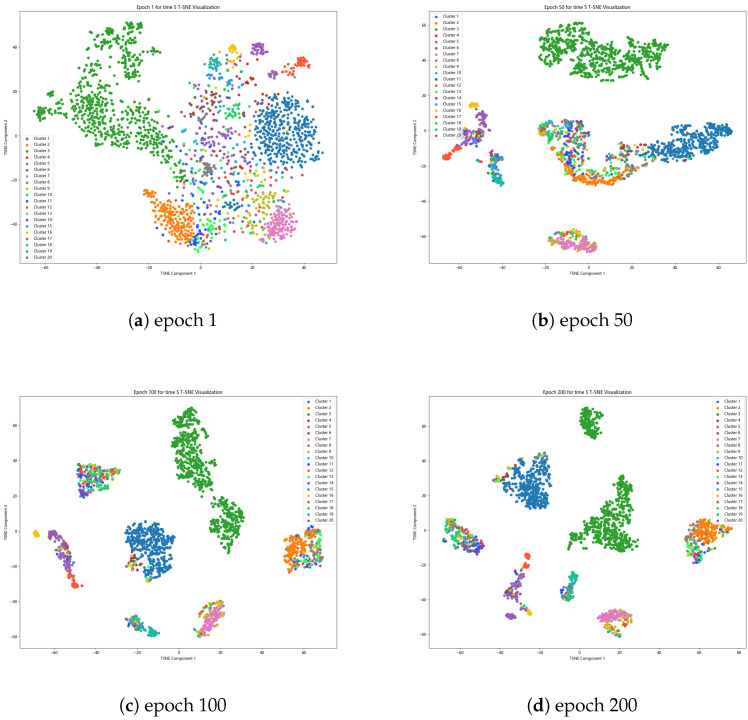
Sensitivity analysis of parameters for Caltech101-20.

**Table 1 entropy-26-00470-t001:** Clustering performance on Caltech101-20, Scene-15, and LandUse-21 datasets.

Methods	Caltech101-20			Scene-15			LandUse-21		
	ACC	NMI	ARI	ACC	NMI	ARI	ACC	NMI	ARI
IMG	44.51	61.35	35.74	24.20	25.64	9.57	16.40	27.11	5.10
EERIMVC	43.28	55.04	30.42	39.60	38.99	22.06	24.92	29.57	12.24
DAIMC	45.48	61.79	32.40	32.09	33.55	17.42	24.35	29.35	10.26
UEAF	47.40	57.90	38.98	34.37	36.69	18.52	23.00	27.05	8.79
DCCAE	44.05	59.12	34.56	36.44	39.78	21.47	15.62	24.41	4.42
PVC	44.91	62.13	35.77	30.83	31.05	14.98	25.22	30.45	11.72
AE2-Nets	49.10	65.38	35.66	36.10	40.39	22.08	24.79	30.36	10.35
DCCA	41.89	59.14	33.39	36.18	38.92	20.87	15.51	23.15	4.43
PICCAE	62.27	67.93	51.56	38.72	40.46	22.12	24.86	29.74	10.48
COMPLETER	70.18	68.06	77.88	41.07	44.68	24.78	25.63	31.73	13.05
ATTENTION	74.88	71.25	86.45	41.93	44.08	25.10	26.68	31.89	13.64
Ours	**77.70**	**73.11**	**92.04**	**44.77**	**45.66**	**26.91**	**27.76**	**33.93**	**13.88**

**Table 2 entropy-26-00470-t002:** Clustering performance on BDGP and MNIST-USPS datasets.

Methods	MNIST-USPS			BDGP		
	ACC	NMI	ARI	ACC	NMI	ARI
DEC	73.10	71.46	63.23	94.78	86.62	87.02
IDEC	76.58	76.89	68.01	95.96	89.40	90.25
BMVC	88.02	89.45	84.48	34.92	12.02	8.33
MVC-LFA	76.78	67.49	60.92	54.68	33.45	28.81
DAMC	71.72	80.85	69.80	98.22	94.61	94.37
SAMVC	69.65	74.58	60.90	53.86	46.25	20.99
CDIMC-net	62.03	67.63	63.38	88.27	78.93	81.94
EAMC	73.04	83.53	72.15	67.56	47.02	39.31
SiMVC	97.74	96.30	95.28	69.72	53.26	44.55
CoMVC	98.47	97.35	98.01	80.68	67.39	59.28
Ours	**99.46**	**98.46**	**98.80**	**99.00**	**96.82**	**97.53**

**Table 3 entropy-26-00470-t003:** Ablative study of main components of the proposed VMC-CD method on Caltech101-20 datasets.

$\ell_{cl}$	$\ell_{ch}$	$\ell_{rec}$	ACC (%)	NMI (%)	ARI (%)
✓	-	✓	68.48	68.11	85.20
✓	✓	-	52.85	56.76	51.62
✓	✓	✓	77.70	73.11	92.04

**Table 4 entropy-26-00470-t004:** Ablative study of main components of the proposed VMC-CD method on Scene-15 datasets.

$\ell_{cl}$	$\ell_{ch}$	$\ell_{rec}$	ACC (%)	NMI (%)	ARI (%)
✓	-	✓	43.79	45.09	26.64
✓	✓	-	41.36	40.06	23.46
✓	✓	✓	44.77	45.66	26.91

**Table 5 entropy-26-00470-t005:** Ablative study of main components of the proposed VMC-CD method on LandUse-21 datasets.

$\ell_{cl}$	$\ell_{ch}$	$\ell_{rec}$	ACC (%)	NMI (%)	ARI (%)
✓	-	✓	27.67	31.12	13.51
✓	✓	-	25.38	29.18	11.97
✓	✓	✓	27.76	33.93	13.88

**Table 6 entropy-26-00470-t006:** Number of iterations and runtime on the aforementioned five datasets.

Dataset	Iterations (epochs)	Running Time (s)
Caltech101-20	200	49.69
Scene-15	500	206.85
LandUse-21	700	148.73
MNIST-UPS	200	71.39
BDGP	200	38.97

## Data Availability

The data presented in this study are available on request from the author.

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
