# Peer review of "View-Driven Multi-View Clustering via Contrastive Double-Learning"

_entropy, 2024, doi:10.3390/e26060470_

Round 1
Reviewer 1 Report
Comments and Suggestions for Authors
1. The authors should thoroughly discuss the methods/algorithms utilized for comparison purposes in Section 2.
2. The authors should address the computational complexity of the algorithm in Section 3.
3. It's better to add a synthetic dataset in the experiment section.
4. The manuscript currently utilizes only three multi-view datasets, which may be insufficient for comprehensive analysis. To enhance the robustness of the study, it is recommended to incorporate two additional multi-view datasets.
5. For a comprehensive evaluation, authors should consider incorporating Fowlkes-Mallows Index (FMI) in addition to the existing evaluation metrics. Including these measures will provide a more holistic assessment of the clustering performance, offering insights into both the similarity of clustering results and the pairwise similarities between clusters.
(i) Fowlkes, E.B. and Mallows, C.L., 1983. A method for comparing two hierarchical clusterings. Journal of the American statistical association, 78(383), pp.553-569.
6. The authors should provide the running time of algorithms in a separate table.
Comments on the Quality of English Language
Moderate editing of English language required.
Author Response
1.The authors should thoroughly discuss the methods/algorithms utilized for comparison purposes in Section 2.
A:Thank you for the reviewer's suggestions.The paper has added a discussion on several recent significant works in Section 2.
2.The authors should address the computational complexity of the algorithm in Section 3.
A:Thank you for the reviewer's suggestions.In the newly added Section 3.7 of this paper, the computational complexity of the algorithm is discussed.
3. It's better to add a synthetic dataset in the experiment section.
A:Thank you for the reviewer's suggestions.In the experimental section, the MNIST-USPS dataset has been included, which is obtained by combining two commonly used handwritten digit recognition datasets.
4.The manuscript currently utilizes only three multi-view datasets, which may be insufficient for comprehensive analysis. To enhance the robustness of the study, it is recommended to incorporate two additional multi-view datasets.
A:Thank you for the reviewer's suggestions.In addition to the aforementioned MNIST-USPS dataset, this paper also incorporates the BDGP dataset. Furthermore, comparisons have been made between this model and other models that perform well on the aforementioned datasets, demonstrating the superior performance and robustness of this model.
5.For a comprehensive evaluation, authors should consider incorporating Fowlkes-Mallows Index (FMI) in addition to the existing evaluation metrics. Including these measures will provide a more holistic assessment of the clustering performance, offering insights into both the similarity of clustering results and the pairwise similarities between clusters.
(i) Fowlkes, E.B. and Mallows, C.L., 1983. A method for comparing two hierarchical clusterings. Journal of the American statistical association, 78(383), pp.553-569.
A:Thank you for the reviewer's suggestions.The FMI metric is utilized to compare the similarity between two clustering results, while the NMI metric employed in this study serves as a normalized indicator of similarity between two clustering results. Although both metrics serve to gauge the similarity of clustering outcomes, they differ slightly in their calculation method and interpretation. It appears that there is some overlap between the two, and opting for a single evaluation metric might be more advantageous.
6.The authors should provide the running time of algorithms in a separate table.
A:Thank you for the reviewer's suggestions.In Section 4.5 of this paper, a separate table is presented, providing the runtime of the algorithm.
Reviewer 2 Report
Comments and Suggestions for Authors
The paper describes an approach for multi-view clustering using a contrastive double-learning method. The authors attempt to perform feature learning from different viewpoints in an attention-driven approach. To that purpose, they make use of contrastive learning at both feature representation and clustering. Empirical results on benchmark datasets showcase the quality of the approach. The paper is well motivated, well written, and addresses an interesting topic. The solution is well formalized and well described. Empirical results are well described and seem satisfactory.
Minor comment: the 3D visual representations in Figure 3 do not seem to clearly convey the results. The authors can consider a revised 2D representation to better showcase the results.
Author Response
Thank you for your suggestion. Due to the presence of two hyperparameters in the loss function of this paper, their combinations need to be explored to uncover the optimal combination of hyperparameters. Additionally, observing how the variations of both hyperparameters affect the performance metrics is crucial. (Possibly 2D representations may not achieve this effect.)
Reviewer 3 Report
Comments and Suggestions for Authors
In this paper,the authors propose a novel multi-view clustering method that introduce a VMC-CD technique, which incorporates valuable information from alternate views while learning feature representations across diverse viewpoints. All in all, this paper is innovative and its presentation is clear. However, there are still some problems to be improved:
1. In Figure 1, contrastive learning modules have confusing lines and spelling errors. There is no subsequent clustering operation after getting the processed data.
2. In Methods, the description of the loss function is a bit sketchy and the author should have spent some ink describing the loss function in detail. There is also no mention of where the processed data goes in the algorithmic flow.
3. In Experimental Setup, the datasets are slightly thin only three need more datasets to show the superiority of the model. The selected comparison methods are more traditional machine learning methods and deep learning methods are older.
4. Because this paper focuses on multi-view clustering problem, several recent important works should be discussed in detail, including “Graph-Collaborated Auto-Encoder Hashing for Multi-view Binary Clustering.”, “Towards Adaptive Consensus Graph: Multi-view Clustering via Graph Collaboration.”, “Tensorial multi-view clustering via low-rank constrained high-order graph learning.” and “Kernelized multiview subspace analysis by self-weighted learning.”
Comments on the Quality of English Language
The Quality of English Language is good .
Author Response
1.In Figure 1, contrastive learning modules have confusing lines and spelling errors. There is no subsequent clustering operation after getting the processed data.
A:Thank you for the reviewer's suggestions.The modifications to Figure 1 have been made, addressing issues with lines and spelling errors. The downstream task of this paper is clustering, thus the purpose of this model is to obtain discriminative feature representations for subsequent clustering operations (as elaborated in the algorithm flow).
2.In Methods, the description of the loss function is a bit sketchy and the author should have spent some ink describing the loss function in detail. There is also no mention of where the processed data goes in the algorithmic flow.
A:Thank you for the reviewer's suggestions.The description of the loss function in the method section has been further elaborated to facilitate reader comprehension. In this section, the paper provides detailed explanations on how the data are processed and how the processed data are utilized to derive the total loss function. The purpose of data processing is to calculate the loss for updating the entire network, thereby obtaining better clustering features.
3. In Experimental Setup, the datasets are slightly thin only three need more datasets to show the superiority of the model. The selected comparison methods are more traditional machine learning methods and deep learning methods are older.
A:Thank you for the reviewer's suggestions.To address the issue of insufficient datasets, this paper has additionally included the MNIST-USPS and BDGP datasets. A comparison has been conducted with datasets that perform well on the aforementioned two datasets, illustrating the robustness of the model proposed in this paper.
4. Because this paper focuses on multi-view clustering problem, several recent important works should be discussed in detail, including “Graph-Collaborated Auto-Encoder Hashing for Multi-view Binary Clustering.”, “Towards Adaptive Consensus Graph: Multi-view Clustering via Graph Collaboration.”, “Tensorial multi-view clustering via low-rank constrained high-order graph learning.” and “Kernelized multiview subspace analysis by self-weighted learning.”
A:Thank you for recommending the paper. The mentioned papers have been incorporated into this study and discussed accordingly.